# Perfusion Patterns in Patients with Chronic Limb-Threatening Ischemia versus Control Patients Using Near-Infrared Fluorescence Imaging with Indocyanine Green

**DOI:** 10.3390/biomedicines9101417

**Published:** 2021-10-09

**Authors:** Pim Van Den Hoven, Lauren N. Goncalves, Paulus H. A. Quax, Catharina S. P. Van Rijswijk, Jan Van Schaik, Abbey Schepers, Alexander L. Vahrmeijer, Jaap F. Hamming, Joost R. Van Der Vorst

**Affiliations:** 1Department of Surgery, Leiden University Medical Center, Albinusdreef 2, 2333 ZA Leiden, The Netherlands; p.van_den_hoven@lumc.nl (P.V.D.H.); laurengoncalves9@gmail.com (L.N.G.); p.h.a.quax@lumc.nl (P.H.A.Q.); j.van_schaik@lumc.nl (J.V.S.); a.schepers@lumc.nl (A.S.); a.l.vahrmeijer@lumc.nl (A.L.V.); j.f.hamming@lumc.nl (J.F.H.); 2Department of Radiology, Leiden University Medical Center, Albinusdreef 2, 2333 ZA Leiden, The Netherlands; c.s.p.van_rijswijk@lumc.nl

**Keywords:** near-infrared, fluorescence imaging, indocyanine green, chronic limb-threatening ischemia, peripheral artery disease, perfusion

## Abstract

In assessing the severity of lower extremity arterial disease (LEAD), physicians rely on clinical judgements supported by conventional measurements of macrovascular blood flow. However, current diagnostic techniques provide no information about regional tissue perfusion and are of limited value in patients with chronic limb-threatening ischemia (CLTI). Near-infrared (NIR) fluorescence imaging using indocyanine green (ICG) has been used extensively in perfusion studies and is a possible modality for tissue perfusion measurement in patients with CLTI. In this prospective cohort study, ICG NIR fluorescence imaging was performed in patients with CLTI and control patients using the Quest Spectrum Platform^®^ (Middenmeer, The Netherlands). The time–intensity curves were analyzed using the Quest Research Framework. Fourteen parameters were extracted. Successful ICG NIR fluorescence imaging was performed in 19 patients with CLTI and in 16 control patients. The time to maximum intensity (seconds) was lower for CLTI patients (90.5 vs. 143.3, *p* = 0.002). For the inflow parameters, the maximum slope, the normalized maximum slope and the ingress rate were all significantly higher in the CLTI group. The inflow parameters observed in patients with CLTI were superior to the control group. Possible explanations for the increased inflow include damage to the regulatory mechanisms of the microcirculation, arterial stiffness, and transcapillary leakage.

## 1. Introduction

Lower-extremity arterial disease (LEAD) is most often caused by atherosclerosis [1,2]. Subsequent hemodynamic alterations leading to hypoxia can trigger a cascade of events leading to macro- and microvascular changes in the affected limb [3]. In the most advanced stage, chronic limb-threatening ischemia (CLTI), blood supply to the lower extremity is insufficient to meet metabolic needs [2,4]. For these patients, a common finding during physical examination of the lower extremities is the appearance of “dependent rubor” or “blanching”, which is presumably caused by dysfunction of the venoarteriolar reflex [5]. In assessing the severity of LEAD, physicians often rely on their clinical judgements of the extremities. The diagnosis is confirmed using conventional measurements of macrovascular blood flow including the ankle-brachial index (ABI), toe pressure measurement, computed tomography (CT) angiography, magnetic resonance angiography, and digital subtraction angiography. However, these techniques provide no information about regional tissue perfusion and have been shown to be of limited value in patients with CLTI [6]. New emerging methods for the assessment of regional tissue perfusion include dynamic volume perfusion CT, laser speckle imaging (LSI), and near-infrared (NIR) fluorescence imaging using indocyanine green (ICG) [7,8,9]. ICG NIR fluorescence imaging has been used in various medical fields for the assessment of tissue perfusion, including cardiac and reconstructive surgery [10,11]. This imaging technique measures fluorescence in the NIR light spectrum (700–1000 nm), which is characterized by low tissue autofluorescence and deep tissue penetration [12]. Upon the intravenous administration of ICG, which has a peak emission of 814 nm, the camera measures the NIR fluorescence intensity over time. The feasibility of ICG as a fluorophore in perfusion assessment can be explained by its confinement to the intravascular compartment due to its ability to bind to plasma proteins [13]. For skin perfusion assessment, ICG NIR fluorescence imaging is currently used intraoperatively in reconstructive surgery to predict flap viability [14]. For patients with LEAD, similar results were seen in predicting skin necrosis following amputation surgery [15]. However, these findings rely on qualitative analyses, meaning that the observer subjectively grades the visualized NIR fluorescence intensity. To quantify and grade of regional tissue perfusion, a better understanding of the different perfusion patterns as observed with ICG NIR fluorescence imaging is needed. Several studies have been performed to quantify ICG NIR fluorescence imaging between patients with different stages of LEAD [16,17,18]. However, inconsistency is seen between stages, and it is unclear whether advanced stages of LEAD alter the in- and outflow of ICG [16]. Furthermore, there is limited information regarding the perfusion patterns of ICG NIR fluorescence imaging in control patients. Therefore, as a first step in the quantification of tissue perfusion using ICG NIR fluorescence imaging, the aim of this study was to analyze the perfusion patterns seen in patients with CLTI and to compare these to non-LEAD control patients.

## 2. Materials and Methods

This prospective cohort study was approved by the Medical Research and Ethics Committee of the Leiden University Medical Center and was registered in the Dutch Trial Register with number NL7531. Patients with CLTI classified according to the global vascular guidelines on the management of CLTI, were included [4]. These were patients who had been diagnosed with either Fontaine stage 3 or stage 4 LEAD. The control group consisted of patients who had undergone intravenous ICG administration prior to liver metastasectomy. Patients were included from December 2018 until April 2021 in a single academic hospital in the Netherlands. Exclusion criteria were allergy or hypersensitivity to sodium iodide, iodide, or ICG; known hyperthyroidism; or autonomous thyroid adenoma, pregnancy, kidney failure (eGFR < 45) and/or severe liver failure. Informed consent was obtained from all patients. ABI and toe pressure measurements were performed in all patients. As an additional measurement for patients with CLTI, duplex ultrasound measurements of the feet were performed, and the highest acceleration in either the dorsalis pedis artery or the posterior tibial artery was reported. These acceleration measurements are described in detail in an earlier study by Brouwers et al. and were performed to assess the severity of arterial stenosis [19]. The Quest Spectrum Platform^®^ (Quest Medical Imaging, Middenmeer, The Netherlands) was used to perform ICG NIR fluorescence imaging (Figure 1). This imaging system is capable of measuring both visible light as well as the NIR signal of ICG. Patients with CLTI were administered an intravenous bolus injection of 0.1 mg/kg ICG (VERDYE 25 mg, Diagnostic Green GmbH, Aschheim-Dornach, Germany) using a peripheral venous line in the cubital fossa or on the dorsum of the hand. Patients in the control group were administered a bolus injection of 10 mg ICG according to local hospital guidelines.

Following the administration of ICG, the NIR fluorescence intensity in both feet was recorded for 10 min (Figure 2). Measurements were performed on patients in a supine position following a rest period of at least 10 min in a room cleared of ambient light. The camera was placed perpendicular to the dorsum of both feet at a distance of 50 cm.

The NIR fluorescence videos were analyzed using the Quest Research Framework^®^ (Version 4.1, Quest Medical Imaging, Middenmeer, the Netherlands). The whole foot was selected as the region of interest (ROI). Upon the selection of the ROI, the software creates a time–intensity curve of the measured intensity in arbitrary units (a.u.). A tracker was used to ensure that the ROI was synchronized with leg movement. Fourteen parameters were extracted from these curves, an explanation of which is given in Figure 3. The ingress rate was defined as the intensity increase per second from baseline to maximum intensity. The Tmax was measured starting at the point of a 10% intensity increase at baseline. The time–intensity curves were also analyzed after normalization for maximum intensity. The curves extracted from these curves were, in percentage per second, the maximum slope ingress and the maximum slope egress. The starting time was defined as an increase of one arbitrary unit for the intensity curves and 1% for the normalized curves. Statistical analyses were performed using IBM SPSS Statistics 25 (IBM Corp. Released 2017 and IBM SPSS Statistics for Windows, Version 25.0. IBM Corp., Armonk, NY, USA). Parameters were compared using the Mann–Whitney U test.

## 3. Results

### 3.1. Patient Characteristics

Successful ICG NIR fluorescence imaging measurements were performed in 35 patients. Nineteen patients presented with LEAD, from whom 28 limbs were classified as CLTI. The control group consisted of 16 patients with a total of 32 limbs. The characteristics for each group are displayed in Table 1. For the CLTI group, 10 limbs were classified as Fontaine stage 4. Compared to the control group, patients in the CLTI group were more likely to present with diabetes, hypertension, and smoking. The mean ABI in the CLTI group was 0.77 versus 1.11 in the control group. The ABI in the CLTI group was not measurable in 9 out of 28 limbs. The acceleration measured on duplex ultrasonography was measured in 22 CLTI limbs with a mean of 0.93 m/s^2^.

### 3.2. ICG NIR Fluorescence Parameters

The results of ICG NIR fluorescence imaging for the 14 extracted parameters are displayed in Table 2.

The mean maximum intensity was significantly lower in the control group (37.9 vs. 25.8 a.u., *p* < 0.001). Furthermore, the time to maximum intensity (i.e., Tmax) was reached earlier in the CLTI group (90.5 vs. 143.3 s, *p* = 0.002). When taking a closer look at the inflow parameters, the maximum slope, the normalized maximum slope, and the ingress rate were all significantly higher in the CLTI group (2.0 vs. 0.6 a.u./s, *p* < 0.001; 4.2 vs. 2.4%/s, *p* < 0.001; 1.0 vs. 0.2 a.u./s, *p* < 0.001). For the outflow parameters, a significant difference was seen for the maximum slope egress, which was higher in the control group (0.5 vs. 0.2 a.u./s, *p* = 0.005). No significant difference was observed for the normalized maximum slope egress (1.0 vs. 0.8%/s, *p* = 0.733). A comparison of the AUC for different intervals following the Tmax displayed no significant difference between the CLTI and control the group.

### 3.3. Time–Intensity Curves

The time–intensity curves for the control group and CLTI group are displayed in Figure 4. Results for the absolute intensity– and the normalized time–intensity curves for both groups are displayed.

Time–intensity curves displaying the absolute intensity change over time show an overall higher absolute intensity for the CLTI group. Following a steep incline in the intensity increase for the CLTI group, the outflow seems comparable with the control patients. The absolute time–intensity curves show a widespread distribution, especially in the CLTI group. In this group, the maximum slope ingress (2.0%/s) has a standard deviation of 2.5 (Table 2). For the AUC egress parameters, standard deviations between 10.0% and 13.5% were observed. When normalizing these time–intensity curves for maximum intensity, both groups display a narrower distribution in all parameters. For the normalized maximum slope in the CLTI group (4.2%/s), a standard deviation of 3.1% was observed. When looking at the AUC egress parameters, the standard deviations had a distribution of 1.8 to 6.1%.

## 4. Discussion

This study demonstrates the different perfusion patterns as seen on ICG NIR fluorescence imaging between patients with CLTI and control patients. Interestingly, most of the inflow parameters observed in patients with CLTI were higher compared to the control group. Concerning the outflow of ICG, however, no significant differences were observed. Furthermore, there was a widespread distribution of measured intensity over time in both groups. There are several earlier studies reporting the use of ICG NIR fluorescence imaging for perfusion assessment in patients with LEAD as well as control patients [7,16,18,20,21,22,23,24,25]. In these studies, an abundance of parameters has been examined, which have been compared to varying diagnosis measurements, including ABI, TP, and transcutaneous oxygen pressure measurements. Patterns of foot perfusion in non-LEAD control patients were analyzed in one study [18]. Regarding inflow parameters, Igari et al. found a prolonged time to maximum intensity for patients with LEAD compared to control patients [18]. No statistical differences were seen for the maximum intensity and T1/2 between the two groups. The differences in the perfusion patterns amongst various stages of LEAD were analyzed in several studies. When comparing inflow parameters between different stages of LEAD, Terasaki et al. observed a prolonged T1/2 for Fontaine stage 3 compared to stage 2; however, this was not observed for stage 4. Regarding outflow, their study concluded that a percentage decrease of 90% in the maximum measured intensity was the most accurate parameter in diagnosing LEAD. For patients with CLTI, Venermo et al. found an increase in the inflow, the PDE10, to be strongly correlated to the transcutaneous oxygen pressure in patients with diabetes mellitus [23]. The same parameter was moderately correlated in patients without diabetes mellitus, suggesting a difference in the perfusion patterns between these groups.

According to the findings in these earlier studies and the results found in this study, the hypothesis that LEAD progression leads to the diminished in- and outflow of ICG is debatable. Several mechanisms might contribute to the increased inflow of ICG seen in patients with CLTI in this study. First, ICG NIR fluorescence imaging is able to penetrate tissue to a depth of several millimeters [26]. Therefore, this imaging technique mainly visualizes the skin with superficial vessels and the superior part of the subcutaneous tissue, i.e., the microcirculation. The nutritional capillaries of this microcirculation in the foot account for approximately 15% of total foot blood flow, which is regulated by various mechanisms, including arteriovenous (AV) shunts [27]. For patients with LEAD and CLTI in particular, this diminished blood flow can lead to hypoxia altering microcirculatory function and can damage these regulatory mechanisms [3,5]. The dysfunction of AV shunts might lead to a relative increase of the blood flow to the skin in patients with CLTI, which also explains the “dependent rubor” seen in this group. Secondly, atherosclerosis leads to stiffness of the arterial wall, which is a common finding in patients with CLTI and that can to an increased pulse wave velocity [28]. In a healthy arterial system, blood flow is gradually transmitted to the peripheral tissue due to the compliance of the vessel wall [29]. This might explain the more gradual perfusion pattern seen in the control group. Furthermore, damage to the microcirculation in CLTI leads to transcapillary leakage, which might further enhance the measured NIR fluorescence intensity. Although a higher dosage of ICG was administered in the majority of patients in the control group, it is unlikely that this would have influenced the perfusion pattern. Moreover, an overall lower absolute intensity was seen in this group. To confirm these findings on increased inflow, a larger cohort of patients with CLTI is needed. Therefore, due to the small sample size, the conclusions in this study must be perceived as a proof of concept. Besides the small cohort size of patients with CLTI, this study is limited by the heterogenous aspect of the CLTI population. In particular, for patients with diabetes mellitus, skin perfusion follows a different pattern than LEAD Fontaine stage 4 patients without diabetes mellitus. Therefore, future studies should distinguish between CLTI patients with and without diabetes mellitus. Furthermore, the control group used in the present study were patients scheduled for liver metastasectomy and therefore might not resemble healthy volunteers in terms of comorbidities. Although LEAD was excluded based on medical history and ABI measurements, there could be differences in the perfusion patterns with healthy volunteers. Therefore, in future patient selection and to further understand perfusion patterns, healthy volunteers should be taken into account as well. With regard to the NIR fluorescence intensity analysis, the use of normalized time–intensity curves seems rational since intensity-related parameters are prone to multiple influencing factors, including camera distance and ICG dosage [30,31]. This normalization minimizes the effect of these influencing factors on the measured intensity and contributes to a narrower distribution, as seen in the time–intensity curves in this study. The use of this normalization might be of use in future research on the quantification of tissue perfusion with ICG NIR fluorescence imaging.

## 5. Conclusions

An increase in the inflow parameters was observed with ICG NIR fluorescence imaging in patients with CLTI compared to control patients. This can possibly be explained by damage to the regulatory mechanisms of microcirculation and arterial stiffness. In order to provide cut-off values for adequate perfusion, more research in lager cohorts is needed on the in- and outflow patterns of control patients and various stages of LEAD.

## Figures and Tables

**Figure 1 biomedicines-09-01417-f001:**
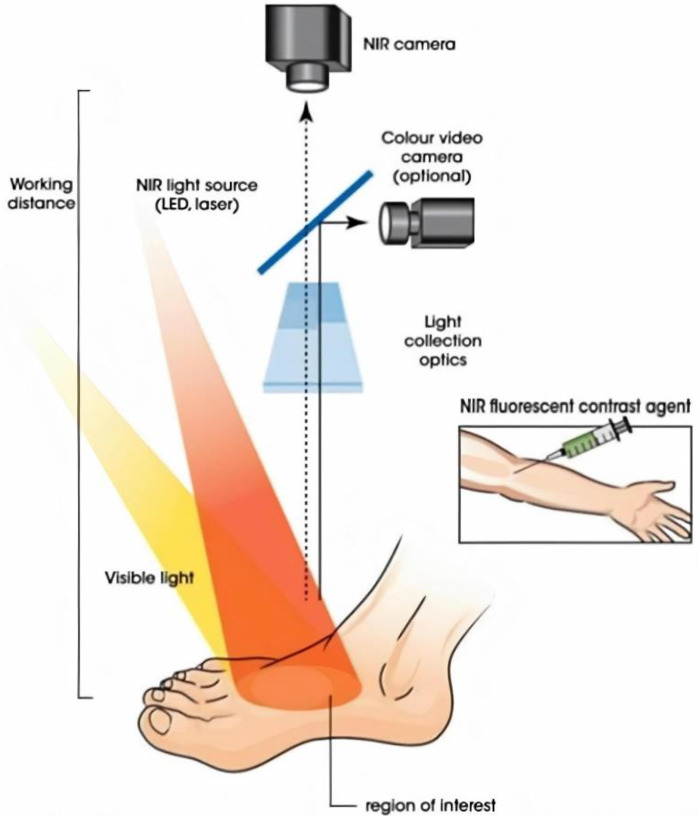
ICG NIR fluorescence imaging setup.

**Figure 2 biomedicines-09-01417-f002:**
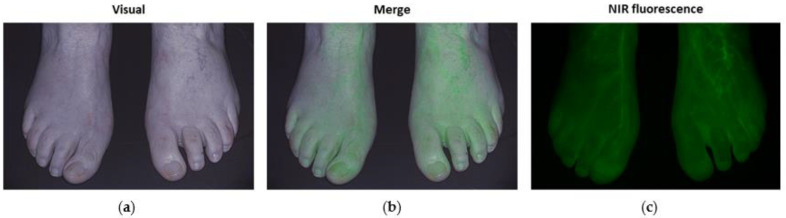
ICG NIR fluorescence imaging in a control patient showing the visual (**a**), merged (**b**), and NIR fluorescence (**c**) output in both feet.

**Figure 3 biomedicines-09-01417-f003:**
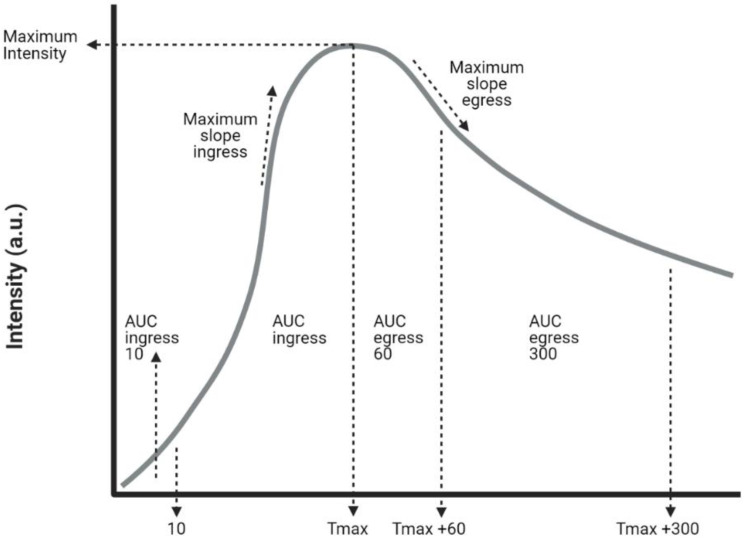
Time–intensity curve with extracted parameters. Abbreviations: a.u, arbitrary unit; AUC, area under the curve.

**Figure 4 biomedicines-09-01417-f004:**
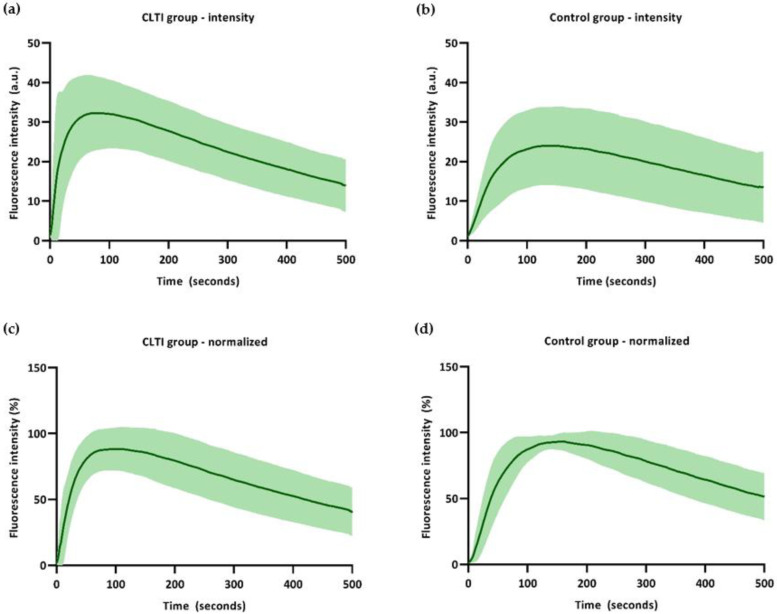
Absolute intensity– and normalized time–intensity curves for the CLTI group and control group: (**a**) Absolute time–intensity curve for the CLTI group; (**b**) absolute time–intensity curve for control group; (**c**) normalized time–intensity curve for the CLTI group; (**d**) normalized time–intensity curve for the control group.

**Table 1 biomedicines-09-01417-t001:** Patient characteristics.

	CLTI	Controls
N (limbs)	19 (28)	16 (32)
Age (SD)	70.4 (7.5)	66.6 (12.3)
Diabetes Mellitus (%)	9 (47.4)	3 (18.8)
Hypertension (%)	15 (78.9)	7 (43.8)
Active smoking (%)	5 (26.3)	1 (6.3)
Fontaine stage limbs, n (%)		
3	18 (64.3)	-
4	10 (35.7)	-
Mean ABI (SD)	0.77 (0.34)	1.11 (0.10)
Mean TP (SD)	44 (25)	106 (22)
Acceleration (SD)	0.93 (1.23)	-

Abbreviations: CLTI, chronic limb-threatening ischemia; SD, standard deviation; ABI, ankle-brachial index; TP, Toe Pressure.

**Table 2 biomedicines-09-01417-t002:** ICG NIR fluorescence imaging parameters.

Parameter	CLTI	Controls	*p*-Value
Maximum intensity (SD)	37.9 (14.4)	25.8 (10.8)	0.000
Maximum slope ingress (SD)	2.0 (2.5)	0.6 (0.4)	0.000
Normalized maximum slope (SD)	4.2 (3.1)	2.4 (1.2)	0.000
Ingress rate (SD)	1.0 (1.7)	0.2 (0.2)	0.000
AUC ingress 10 (SD)	47.4 (2.2)	48.8 (3.3)	0.073
AUC ingress (SD)	71.4 (6.3)	70.6 (3.8)	0.213
Tmax (SD)	90.5 (53.4)	143.3 (64.5)	0.002
Maximum slope egress (SD)	0.5 (0.7)	0.2 (0.1)	0.005
Normalized maximum slope egress (SD)	1.0 (0.9)	0.8 (0.3)	0.733
AUC egress 60 (SD)	92.8 (10.0)	96.7 (1.8)	0.113
AUC egress 120 (SD)	87.9 (11.9)	92.8 (2.3)	0.127
AUC egress 180 (SD)	82.9 (12.9)	88.3 (4.3)	0.164
AUC egress 240 (SD)	78.2 (13.3)	83.8 (5.4)	0.168
AUC egress 300 (SD)	73.7 (13.5)	73.3 (6.0)	0.271

Abbreviations: SD, standard deviation; CLTI, chronic limb-threatening ischemia; AUC, area under the curve.

## Data Availability

The data presented in this study are available upon request from the corresponding author.

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
