# Peer review of "Perfusion Patterns in Patients with Chronic Limb-Threatening Ischemia versus Control Patients Using Near-Infrared Fluorescence Imaging with Indocyanine Green"

_biomedicines, 2021, doi:10.3390/biomedicines9101417_

Round 1

Reviewer 1 Report

The authors must explain why they have chosen this specific control group and not a healthy control group

The main limitation is the RF of patients included: mainly RF 3 and for CLTI only RF 4 and not 5 or 6

Author Response

Point 1: The authors must explain why they have chosen this specific control group and not a healthy control group

Response 1: We agree with the reviewer that this is a limitation of our control group. This specific control group was chosen because these patients were already administered Indocyanine Green one day prior to liver surgery as part of the protocol. Following exclusion of the presence of lower extremity arterial disease by medical history, supported by a normal ankle brachial index, we believed this control group would be feasible, without having to harm healthy volunteers. However, due to the presence of comorbidities in this control group, differences in near-infrared fluorescence imaging perfusion patterns could indeed be observed when these are compared with healthy volunteers. We added a section to the discussion regarding this matter and addressed the need to assess near-infrared fluorescence imaging in healthy volunteers as well (page 7, line 52 to page 8, line 4, see below). We hope this sufficiently addresses the reviewer’s comment.

Adjustments:

Page 7, line 52 to page 8, line 4: Furthermore, the control group used in the present study were patients scheduled for liver metastasectomy and therefore might not resemble healthy volunteers in terms of comorbidities. Although LEAD was excluded based on medical history and ABI measurements, there could be differences in perfusion patterns with healthy volunteers. Therefore, in future patient selection and further understanding perfusion patterns, healthy volunteers should be taken into account as well.

Point 2: The main limitation is the RF of patients included: mainly RF 3 and for CLTI only RF 4 and not 5 or 6

Response 2: We understand the reviewers comment, since we did not clearly state in our “Materials and Methods” that we used the Fontaine stage classification instead of the Rutherford classification. We included patients with Fontaine stage 3 and 4, defined as CLTI, which correspond to Rutherford stages 4,5 and 6 as suggested by the reviewer. We added this to the Methods section (page 2, line 29-30, see below). We thank the reviewer for this valuable comment.

Adjustments:

Page 2, line 29-30: Patients with CLTI as classified according to the Global Vascular Guidelines on the management of CLTI, were included [19]. These were patients diagnosed with either Fontaine stage 3 or stage 4 LEAD.

Reviewer 2 Report

The authors present an interesting manuscript on NIR fluorescence imaging with indocyanine green in patients with chronic limb-threatening ischemia.

The aim of the study is clearly named, the introduction is expediently written. The methods are described well and in detail. The data are presented clearly. However, only a small number of patients with and without CLTI were recruited for the study during the 2.5 years of enrollment. A bigger number of patients would give the data more significance so that this study can only be a proof of concept - this should be stated in the manuscript.

In addition, the authors should discuss the ICG imaging in some more detail. In the last ten years several study on this technique were performed in patients with peripheral artery disease. A more detailed discussion of previously gained information, the use of ICG and the now added data in context of the existing studies would increase the value of this manuscript.

The 'acceleration measured on duplex ... with a mean of 0.93 m/s2.' confuses. What do the authors mean? Where did they perform duplex ultrasound? What do they want to express with this parameter?

As a minor aspect, the resolution of figures 1 and 4 needs to be improved.

Author Response

Point 1: The authors present an interesting manuscript on NIR fluorescence imaging with indocyanine green in patients with chronic limb-threatening ischemia. The aim of the study is clearly named, the introduction is expediently written. The methods are described well and in detail. The data are presented clearly. However, only a small number of patients with and without CLTI were recruited for the study during the 2.5 years of enrollment. A bigger number of patients would give the data more significance so that this study can only be a proof of concept - this should be stated in the manuscript.

Response 1: We want to thank the reviewer for the compliments on our manuscript. We agree with the reviewer that only a small number of patients was used which precludes fierce statements on differences in perfusion patterns. We added a section to the discussion in which we emphasized the interpretation of the results as a proof of concept (page 7, lines 46-48). Furthermore, we emphasized the need for larger cohorts in future studies in the conclusion (page 8, line 15).

Point 2: In addition, the authors should discuss the ICG imaging in some more detail. In the last ten years several study on this technique were performed in patients with peripheral artery disease. A more detailed discussion of previously gained information, the use of ICG and the now added data in context of the existing studies would increase the value of this manuscript.

Response 2: We agree with the reviewer on this matter and described the earlier use of this technique in patients with peripheral artery disease in a more detailed matter in the discussion (page 7, lines 7-24). Furthermore, in this section we underlined the fact that a control group was described in only one study. We hope to have sufficiently addressed this reviewer’s comment by adding this section to the discussion.

Point 3: The 'acceleration measured on duplex ... with a mean of 0.93 m/s2.' confuses. What do the authors mean? Where did they perform duplex ultrasound? What do they want to express with this parameter?

Response 3: We understand the reviewers comment and wish to clarify the use of this measurement. We performed duplex ultrasound of the feet in patients with CLTI and reported the highest measured acceleration in either the dorsalis pedis artery or posterior tibial artery. This acceleration, measured in the upslope segment of the duplex ultrasound wave, is a technique to describe the severity of an arterial stenosis, which is described in detail in an earlier study by Brouwers et al. published in the Journal of Vascular Surgery. We added a section to the “Materials and methods” section to clarify the report of this measurement (page 2, lines 36-41). We hope to have provided the reviewer with a sufficient answer regarding this matter.

Point 4: As a minor aspect, the resolution of figures 1 and 4 needs to be improved.

Response 4: We have improved the resolution of these figures.

Round 2

Reviewer 1 Report

The authors addressed the concerns of the reviewers in an adequate manner